# Bone Marrow Aspirate Concentrate versus Human Umbilical Cord Blood-Derived Mesenchymal Stem Cells for Combined Cartilage Regeneration Procedure in Patients Undergoing High Tibial Osteotomy: A Systematic Review and Meta-Analysis

**DOI:** 10.3390/medicina59030634

**Published:** 2023-03-22

**Authors:** Dojoon Park, Youn Ho Choi, Se Hyun Kang, Hae Seok Koh, Yong In

**Affiliations:** 1Department of Orthopedic Surgery, St. Vincent Hospital, College of Medicine, 93, Jungbu-daero, Paldal-gu, Suwon-si 16247, Republic of Korea; onedream1106@naver.com (D.P.); kkieek@daum.net (Y.H.C.); seihyunk@naver.com (S.H.K.); vincentos@naver.com (H.S.K.); 2Department of Orthopaedic Surgery, Seoul St. Mary’s Hospital, College of Medicine, The Catholic University of Korea, 222, Banpo-daero, Seocho-gu, Seoul 06591, Republic of Korea

**Keywords:** high tibial osteotomy, bone marrow aspirate concentrate, human umbilical cord blood-derived mesenchymal stem cells, knee osteoarthritis

## Abstract

*Background and objectives:* Cartilage regeneration using mesenchymal stem cells (MSCs) has been attempted to improve articular cartilage regeneration in varus knee osteoarthritis (OA) patients undergoing high tibial osteotomy (HTO). Bone marrow aspirate concentrate (BMAC) and human umbilical cord blood-derived MSCs (hUCB-MSCs) have been reported to be effective. However, whether BMAC is superior to hUCB-MSCs remains unclear. This systematic review and meta-analysis aimed to determine the clinical efficacy of cartilage repair procedures with BMAC or hUCB-MSCs in patients undergoing HTO. *Materials and Methods:* A systematic search was conducted using three global databases, PubMed, EMBASE, and the Cochrane Library, for studies in which the clinical outcomes after BMAC or hUCB-MSCs were used in patients undergoing HTO for varus knee OA. Data extraction, quality control, and meta-analysis were performed. To compare the clinical efficacy of BMAC and hUCB-MSCs, reported clinical outcome assessments and second-look arthroscopic findings were analyzed using standardized mean differences (SMDs) with 95% confidence intervals (CIs). *Results:* The present review included seven studies of 499 patients who received either BMAC (BMAC group, *n* = 169) or hUCB-MSCs (hUCB-MSC group, *n* = 330). Improved clinical outcomes were found in both BMAC and hUCB-MSC groups; however, a significant difference was not observed between procedures (International Knee Documentation Committee score; *p* = 0.91, Western Ontario and McMaster Universities OA Index; *p* = 0.05, Knee Society Score (KSS) Pain; *p* = 0.85, KSS Function; *p* = 0.37). On second-look arthroscopy, the hUCB-MSC group showed better International Cartilage Repair Society Cartilage Repair Assessment grade compared with the BMAC group (*p* < 0.001). *Conclusions:* Both BMAC and hUCB-MSCs with HTO improved clinical outcomes in varus knee OA patients, and there was no difference in clinical outcomes between them. However, hUCB-MSCs were more effective in articular cartilage regeneration than BMAC augmentation.

## 1. Introduction

High tibial osteotomy (HTO) is a reliable surgical method for physically active or young patients with medial compartment knee osteoarthritis (OA) with varus deformity [1,2,3]. The HTO procedure reduces the weight loading of the medial compartment through alteration of the weight-bearing axis, and healing of the damaged cartilage can be expected due to reduced stress [4,5]. Satisfactory clinical and radiological improvement after HTO at short-term and mid-term follow-up have been reported in several studies [6,7]; however, long-term follow-up data revealed a troubling trend of deteriorating outcomes in some patients, necessitating revision total knee arthroplasty (TKA) due to unsatisfactory results [8,9]. Therefore, a combination of additional cartilage regeneration procedures with HTO has been attempted to improve articular cartilage regeneration and surgical outcomes [10]. Among additional procedures, mesenchymal stem cell (MSC) enhancement is an emerging option for cartilage regeneration procedures for OA [11,12], and its effectiveness has been demonstrated in several clinical trials [13,14,15].

Bone marrow aspirate concentrate (BMAC) and human umbilical cord blood-derived mesenchymal stem cells (hUCB-MSCs) are representative stem cell-based orthobiologics [16]. BMAC augmentation relies on the inclusion of many growth factors and pluripotent stromal cells that induce MSC differentiation into chondrocytes, potentially leading to the production of native, hyaline-like cartilage [17,18]. hUCB-MSCs have low immunogenicity and can be collected using a noninvasive method. In addition, hUCB-MSCs have a good expansion capacity to provide sufficient cells for treatment [19,20].

The superior results of the combination of HTO and cartilage regeneration procedures compared with HTO alone have been reported in several studies [21,22]. Despite promising results regarding cartilage regeneration procedures using MSC-based orthobiologics, the optimal cartilage regeneration procedure that can be performed with HTO remains unclear. Direct comparison of the efficacy of BMAC or hUCB-MSCs combined with HTO has been conducted in few studies to date [23,24].

The purpose of this meta-analysis was to identify available studies on the clinical efficacy of HTO with cartilage regeneration procedure using BMAC or hUCB-MSCs in varus knee OA patients, and to compare the efficacy of the two procedures for clinical improvement and cartilage regeneration. To the best of our knowledge, this is the first meta-analysis in which this topic was investigated. We hypothesized equivalent clinical effects of BMAC and hUCB-MSCs with HTO in varus knee OA patients.

## 2. Materials and Methods

### 2.1. Meta-Analyses Principles

This analysis was conducted under the Preferred Reporting Items for Systematic Reviews and Meta-Analysis (PRISMA) principle [25]. Studies in which the clinical effects of BMAC and hUCB-MSCs in patients undergoing HTO for varus knee OA were systematically reviewed. Ethical approval was unnecessary because all analyses were conducted using existing literature.

### 2.2. Search Strategy

A systematic search was conducted by two reviewers using three global online databases (PubMed, EMBASE, and Cochrane Library) for studies published by 22 September 2022, in which clinical effects after the use of BMAC or hUCB-MSCs in patients undergoing HTO for varus knee OA were investigated. The publication language was restricted to English.

A search for relevant articles was conducted using various combinations of the following keywords: “osteoarthritis”, “knee”, “high tibial osteotomy”, “bone marrow aspirate concentrate”, and “umbilical cord blood-derived mesenchymal stem cells.” Details of the search terms and strategy are presented in Appendix A.

A secondary manual search was performed on the related reviews and meta-analyses and their reference lists to retrieve relevant articles that were not identified using the databases.

### 2.3. Study Criteria and Screening Process

Two independent reviewers evaluated the eligibility of potentially relevant articles retrieved after removing duplicates based on the predefined criteria. The review was conducted on the title and abstract of the study and, in cases of uncertainty, was performed on the entire text. Discrepancies were discussed with a third reviewer.

The inclusion criteria were as follows: (1) studies using BMAC or hUCB-MSCs with HTO for treatment of varus knee OA, (2) studies investigating the clinical effects of BMAC or hUCB-MSCs with HTO on postoperative outcome, and (3) studies with a mean follow-up period >18 months. The exclusion criteria were as follows: (1) reviews, meta-analyses, case reports, and letters; (2) duplicates of previously published articles.

### 2.4. Data Extraction and Quality Control

Data extraction was performed using standardized protocols. The following variables of the included studies were collected: first author, publication year, study design, level of evidence, type of osteotomy, type of intervention, sex, age, body mass index (BMI), sample size, preoperative Kellgren–Lawrence grade [26], preoperative International Cartilage Repair Society (ICRS) grade [27], cartilage defect size, mean follow-up duration, preoperative hip-knee-ankle (HKA) angle, clinical assessment, postoperative ICRS-Cartilage Repair Assessment (CRA) grade [28], postoperative Koshino stage [29], a brief description of the preparation method of intervention, postoperative rehabilitation protocol, and reported adverse events.

Because the final eligible studies did not include randomized controlled trials, two independent reviewers evaluated the bias and risk of all eligible observational studies using the Methodological Item for Non-Randomized Studies (MINORS) [30], with 12 categories for comparative studies and 8 categories for noncomparative studies. Each category was rated 0 (not reported), 1 (reported but inadequate), or 2 (reported and adequate).

### 2.5. Statistical Analyses

Descriptive statistics including the mean and standard deviation for any numerical variable were recorded. Any missing standard deviation was estimated using pre-established methodologies [31]. The 95% confidence interval (CI) of the appropriate variable was reported. Heterogeneity was determined using the I^2^ statistic; if I^2^ < 50% (low heterogeneity), a fixed-effects model was used; otherwise, a random-effects model was used, and a funnel plot was used to assess the existence of publication bias (Appendix A). A *p*-value < 0.05 was considered statistically significant.

Clinical outcome was evaluated using the mean International Knee Documentation Committee (IKDC) score [32], Western Ontario and McMaster Universities OA Index (WOMAC) score [33], and other clinical outcome measures performed before and after surgery. The quality of articular cartilage regeneration was determined using the ICRS-CRA on second-look arthroscopy. Clinical efficacy based on the type of treatment was estimated using the standardized mean difference (SMD) and was analyzed to compare the clinical effects between patients who received BMAC (BMAC group, *n* = 169) or hUCB-MSCs (hUCB-MSC group, *n* = 330). Statistical significance was set at *p* < 0.05. All statistical analyses were conducted using Review Manager software (version 5.3; The Nordic Cochrane Centre, Copenhagen, Denmark).

## 3. Results

### 3.1. Study Characteristics

Details of the search process, including literature identification and verification, are presented in Figure 1. A total of 33 potentially relevant studies were retrieved from 3 databases, of which 23 remained after 10 duplicate articles were removed. During the review process, the titles and abstracts of the studies were reviewed, and for three uncertain cases, the review was extended to include full texts for identification. One article was added after a related review article search. After comprehensive screening, 7 studies involving 499 patients were included in this systematic review.

The outcomes of HTO with BMAC were reported in two studies, the outcomes of HTO with hUCB-MSCs were reported in three studies, and comparative outcomes between HTO with BMAC and HTO with hUCB-MSCs were reported in two studies. The baseline characteristics of the studies are summarized in Table 1, Table 2 and Table 3.

### 3.2. Methodological Quality

The level of evidence was 3 in three studies [23,24,34] (42.9%) and 4 in four studies [35,36,37,38] (57.1%). The mean MINORS score for comparative studies (three studies, maximum of 24 points) was 14.7 ± 2.4 (range 13–18), and noncomparative studies (four studies, maximum of 16 points) had a mean MINORS score of 9 ± 1.5 (range, 6–11; Appendix A).

**Table 1 medicina-59-00634-t001:** Characteristics of the included studies.

Study	Level of Evidence	Design	Type of Osteotomy	Intervention	Gender (M, F)	Age (Years)	BMI (kg/m^2^)	Number of Patients in Intervention Group
Cavallo, 2018 [35]	IV	retrospective case series study	HTO (medial opening-wedge)	bone marrow-derived cells + PRF	BMAC (15, 9)	47.9 ± 12.3	Not reported	24
Song, 2020 [36]	IV	retrospective case series study	HTO	hUCB-MSC + multiple holes	hUCB-MSC (2, 23)	64.9 ± 4.4	24.9 ± 3.1	25
Song, 2020 [37]	IV	retrospective case series study	HTO (uniplanar osteotomy)	hUCB-MSC + multiple holes	hUCB-MSC (30, 95)	58.3 ± 6.8	25.6 ± 2.7	125
Jin, 2021 [34]	III	retrospective comparative study	HTO (biplanar opening wedge osteotomy)	BMAC + MFx vs. MFx	BMAC (11, 37)	56.9 ± 6.1	25.8 ± 3.1 (range, 18.1–33.2)	48
Chung, 2021 [38]	IV	retrospective case series study	HTO (biplanar opening wedge osteotomy)	hUCB-MSC + multiple drill holes	Not reported	56.6 (range, 43–65)	25.8 (range, 20.9–33.2)	93
Lee, 2021 [24]	III	retrospective comparative study	HTO (biplanar opening wedge osteotomy)	BMAC + MFx vs. hUCB-MSC + MFx	BMAC (6, 36); hUCB-MSC (6, 26)	BMAC, 60.7 ± 4.1; hUCB-MSC, 58.1 ± 3.6	BMAC, 26.1 ± 2.8; hUCB-MSC, 26.6 ± 3	BMAC (42), hUCB-MSC (32)
Yang, 2022 [23]	III	retrospective cohort study	HTO (biplanar opening wedge osteotomy)	BMAC + MFx vs. hUCB-MSC + MFx	BMAC (17, 38); hUCB-MSC (13, 42)	BMAC, 55.0 ± 7.3; hUCB-MSC, 56.4 ± 5.3	BMAC, 27.2 ± 3.9 hUCB-MSC, 26.8 ± 3.2	BMAC (55), hUCB-MSC (55)

BMI, body mass index; Pre-OP, preoperative; K-L, Kellgren–Lawrence; HTO, high tibial osteotomy; PRF, platelet-rich fibrin; hUCB-MSCs, human umbilical cord blood-derived mesenchymal stem cells; BMAC, bone marrow aspirate concentrate; MFx, microfracture.

**Table 2 medicina-59-00634-t002:** Preoperative status and outcomes of interventions of the included studies.

Study	Pre-OP K-L Grade	Pre-OP ICRS Grade	Defect Size (MFC, cm^2^)	Mean Follow-Up (Months)	HKA Angle (Pre-OP vs. Post-OP)	Clinical Assessment	Number of Patients Undergoing 2nd-Look Arthroscopy	Post-OP ICRS-CRA Grade	Post-OP Koshino Staging
Cavallo, 2018 [35]	IV or less	Not reported	Not reported	44.4 ± 17.7	Pre-OP, varus 1–15°; Post-OP, Not reported	IKDC, KOOS, VAS, Tegner	NA	Not reported	Not reported
Song,2020 [36]	Not reported	Not reported	7.2 ± 1.9	26.7 ± 1.8	Pre-OP, ≥3°; Post-OP, Not reported	IKDC, VAS, WOMAC	14	I (6), II (8), III (0), IV (0)	Not reported
Song, 2020 [37]	III or less	IV (125)	6.9 ± 2.0	Not reported	Pre-OP, 7.6 ± 2.4; Post-OP, Not reported	IKDC, WOMAC, VAS	125	I (73), II (37), III (15), IV (0)	Not reported
Jin, 2021 [34]	III (36), IV (12)	III (41), IV (7)	2.3 ± 0.9	33.6 ± 6.6	7.5 ± 3.4 vs. −2.9 ± 2.5	IKDC, WOMAC, KSS-pain, KSS-function	33	I (1), II (18), III (11), IV (3)	A (2), B (15), C (16)
Chung, 2021 [38]	III	III or IV	6.5 (range, 2.0–12.8)	20.4 (range, 12–42)	Pre-OP, >3°; Post-OP, Not reported	IKDC, WOMAC, KSS-pain, KSS-function, HSS	49	I (4), II (34), III (11), IV (0)	A (0), B (12), C (37)
Lee, 2021 [24]	Not reported	BMAC, MFC 3.9 ± 0.3, MTC 3.9 ± 0.3; hUCB-MSC, MFC 3.9 ± 0.3, MTC 3.9 ± 0.3	BMAC, 6.5 ± 2.9; hUCB-MSC, 7 ± 1.9	BMAC, 20.7 ± 6.1; hUCB-MSC, 15.6 ± 2.8	BMAC, 8.6 ± 3.1 vs. −2.8 ± 3.2; hUCB-MSC, 7.4 ± 2.6 vs. −2.9 ± 1.6	HSS, WOMAC, KSS-pain, KSS-function	BMAC (42), hUCB-MSC (32)	BMAC (42), I (1), II (18), III (12), IV (11); hUCB-MSC (32), I (6), II (20), III (6), IV (0)	Not reported
Yang, 2022 [23]	III	BMAC (55), III (5), IV (50); hUCB-MSC (55), III (3), IV (52)	BMAC, 6.4 ± 3.1; hUCB-MSC, 6.2 ± 2.4	BMAC, 34.2 ± 8.4; hUCB-MSC, 31.0 ± 6.0	BMAC, 7.6 ± 2.9 vs. −1.5 ± 2.3; hUCB-MSC, 7.5 ± 2.7 vs. −1.6 ± 2.2	IKDC, KOOS, SF-36, Tegner	BMAC (37), hUCB-MSC (44)	BMAC (37), I (1), II (20), III (11), IV (5); hUCB-MSC (44), I (4), II (30), III (10), IV (0)	BMAC (37), A (4), B (12), C (21); hUCB-MSC (44), A (0), B (12), C (32)

Pre-OP, preoperative; K-L, Kellgren–Lawrence; hUCB-MSCs, human umbilical cord blood-derived mesenchymal stem cells; BMAC, bone marrow aspirate concentrate; MFC, medial femoral condyle; HKA, hip–knee–ankle; IKDC, International Knee Documentation Committee; KOOS, Knee injury and Osteoarthritis Outcome Score; VAS, Visual Analog Scale; WOMAC, Western Ontario and McMaster Universities Arthritis index; KSS, Knee Society Score; ICRS-CRA, International Cartilage Repair Society Cartilage Repair Assessment; SF-36, Short Form 36 Health Survey.

**Table 3 medicina-59-00634-t003:** Preoperative preservation, rehabilitation and complications of the included studies.

Study	Pre-OP BMAC or hUCB-MSC Preservation	Rehabilitation	Complication	Number of Patients in Control Group
Cavallo, 2018 [35]	60 mL of bone marrow from the posterior iliac crest, 15 min centrifugation cycles, a collagen scaffold	Partial weight-bearing was after six weeks; full weight-bearing at eight weeks after evaluation of bone consolidation	Knee swelling due to hemarthrosis in three cases; two cases of infrapatellar nerve injury; one case of delayed union of the osteotomy	NA
Song, 2020 [36]	hUCB-MSCs were used as a stem cell drug (CARTISTEM, MEDIPost-OP), mixed with sodium hyaluronate, therapeutic dosage 500 μL/cm^2^ of the defect area with a cell concentration of 0.5 × 10^7^ cells/mL	Non-weight-bearing for eight weeks, full weight-bearing after twelve weeks	Not reported	NA
Song, 2020 [37]	CARTISTEM^®^ (MediPost-OP, Seongnam-si, Gyeonggi-do, Republic of Korea), 1.5 mL hUCB-MSCs (7.5 × 10^6^ cells/vial) and 4% hyaluronic acid (HA) hydrogel, therapeutic dose 500 mL/cm^2^	Partial weight-bearing after four weeks; full weight-bearing at six weeks	Not reported	NA
Jin, 2021 [34]	contralateral anterior superior iliac spine, at least 40 mL of bone marrow, centrifuged for 4 min at 2500 rpm	Non-weight-bearing to partial weight-bearing after six weeks; full weight-bearing after twelve weeks considering bone healing	Not reported	MFx (43)
Chung, 2021 [38]	Cartistem, 1.5 mL of cord blood-derived MSCs (7.5 × 10^6^) and 4% HA, 500 mL/cm^2^ of defect with a cell concentration of 0.5 × 10^7^ cells/mL	Non-weight-bearing to partial weight-bearing walking after six weeks; full weight-bearing after twelve weeks based on the level of bone healing	Some patients, knee swelling for up to one month	NA
Lee, 2021 [24]	BMAC, contralateral anterior superior iliac spine, at least 40 mL of bone marrow, centrifuge for 4 min at 2500 rpm; Cartistem, hUCB-MSCs-HA hydrogel composites	Non-weight-bearing to partial weight-bearing after six weeks; full weight-bearing after twelve weeks depending on the level of bone healing	Not reported	NA
Yang, 2022 [23]	BMAC, contralateral anterior superior iliac spine, at least 40 mL of bone marrow, centrifuged for 4 min at 2500 rpm; Cartistem, 1.5 mL of cord blood-derived MSCs (7.5 × 10^6^) and 4% HA, 500 mL/cm2 of defect with a cell concentration of 0.5 × 10^7^ cells/mL	Partial weight-bearing after six weeks; full weight-bearing after twelve weeks.	BMAC: one patient complained of postoperative stiffness	NA

Pre-OP, preoperative; HTO, hUCB-MSCs, human umbilical cord blood-derived mesenchymal stem cells; BMAC, bone marrow aspirate concentrate; HA, hyaluronic acid; NA, not applicable.

### 3.3. Meta-Analysis Results

The study population consisted of 499 patients treated with HTO and either BMAC (169 patients) or hUCB-MSCs (330 patients). The mean age of subjects was not significantly different between the BMAC and hUCB-MSC groups (57.7 ± 7.3 and 57.8 ± 5.4 years, respectively; *p* = 0.86). In all studies except one (6/7; 85.7%), preoperative BMI was reported. Based on the available data, the BMAC group had a higher BMI than the hUCB-MSC group (26.3 ± 3.4 vs. 25.9 ± 2.7 kg/m^2^, *p* = 0.02).

#### 3.3.1. Clinical Outcomes

**IKDC**: In six [23,34,35,36,37,38] of the seven studies, the IKDC score used to evaluate the clinical effects of BMAC or hUCB-MSCs with HTO in patients with varus knee OA was reported. The IKDC subgroup consisted of 127 patients treated using BMAC, and 298 patients treated using hUCB-MSCs. Significant heterogeneity was found (*p* < 0.001, I^2^ = 94%), and the random effects model was used. Patients treated with BMAC had significantly improved IKDC scores (SMD, 4.13; 95% CI, 1.23–7.00), as did those in the hUCB-MSC group (SMD, 3.92; 95% CI, 3.65–4.20). However, based on IKDC score, the clinical effects of BMAC vs. hUCB-MSCs for combined cartilage regeneration in patients undergoing HTO were equivalent (*p* = 0.91; Figure 2).

**WOMAC**: In 5 studies [24,34,36,37,38], the WOMAC score was reported, and 90 and 275 participants were included in the BMAC and hUCB-MSC groups, respectively. Since only one [38] of five studies reported WOMAC subscales, while the other four reported only a total WOMAC score, and only the total WOMAC score was compared. Significant heterogeneity was found (*p* < 0.001; I^2^ = 93%), and a random effects model was used. Based on the WOMAC index, both BMAC and hUCB-MSCs had a significant clinical effect compared with the preoperative status (SMD 2.09, 95% CI, 1.25–2.93 for BMAC and 3.39, 95% CI, 2.39–4.397 for hUCB-MSCs). However, the difference between groups did not reach statistical significance (*p* = 0.05; Figure 3).

**Other reported clinical outcomes**: Other reported outcomes were KSS score (three studies), VAS (three studies), KOOS (two studies), HSS (two studies), SF-36 (one study), and Tegner activity score (two studies). The mean preoperative and postoperative values for the outcome scales are shown in Table 4. KSS was the only measure used in more than one study to evaluate BMAC and hUCB-MSCs; it was used in three studies [24,34,38]. Mean KSS subscale (pain and function) values were reported in all studies; statistically significant heterogeneity was detected (KSS pain, *p* = 0.01, I^2^ = 73%; KSS function, *p* < 0.001, I^2^ = 84%). In a random effects model, patients treated with BMAC or hUCB-MSCs showed improved clinical outcome after surgery (KSS pain SMD, 1.51, 95% CI, 0.43–2.59 for BMAC and 1.40; 95% CI, 1.12–1.68 for hUCB-MSCs; KSS function SMD, 1.99, 95% CI, 0.63–3.34 for BMAC and 1.35; 95% CI, 1.08–1.63 for hUCB-MSCs). However, differences in KSS pain and function scores were not found between the two treatment groups (*p* = 0.85 and *p* = 0.37, respectively; Figure 4).

#### 3.3.2. Second-Look Arthroscopic Findings

In six studies [23,24,34,36,37,38], second-look findings were reported, and four studies [23,24,34,37] compared preoperative ICRS grade with postoperative ICRS-CRA grade. Significant heterogeneity was found, and a random effects model was used. Greater improvement in articular cartilage regeneration was observed in patients treated with hUCB-MSCs (SMD, 4.18; 95% CI, 3.61–4.75) than in patients treated with BMAC (SMD, 1.81; 95% CI, 1.10–2.53; *p* < 0.001, Figure 5).

## 4. Discussion

The main finding of this study was that patients who underwent HTO for varus knee OA had improved clinical outcomes regardless of whether BMAC or hUCB-MSCs were used for cartilage regeneration. Postoperative clinical outcomes were significantly improved in both groups compared with preoperative baseline measurements, and both treatments provided reliable results in terms of functional score improvement and pain relief, without group differences. However, in comparison of second-look arthroscopic findings, hUCB-MSCs were more effective in articular cartilage regeneration than BMAC.

HTO provides a mechanical environment that can prevent the progression of degenerative changes by correcting the axis in which the weight is focused on the medial side of the knee of OA patients. Several studies have reported successful short- and medium-term outcomes after HTO [6,7]. However, with long-term follow-up, a trend for worsening outcomes was observed, with some patients requiring revision TKA due to poor outcomes [8,9]. It is unclear if deterioration of HTO results over time is associated with inappropriate cartilage regeneration of medial osteoarthritic cartilage [4,39].

Therefore, many attempts have been made to combine HTO with several types of cartilage regeneration procedures to improve long-term outcomes. Among such procedures, augmentation using MSCs that can differentiate into chondrocytes and produce extracellular matrix molecules important for cartilage regeneration is a promising option for managing cartilage defects [40]. Because the etiology of knee OA involves both biomechanical and biochemical changes in knee articular cartilage, combining HTO and MSC augmentation procedures appears promising.

MSCs concentrated after extraction from autologous bone marrow were introduced as a next-generation therapy for cartilage disease [34]. This procedure has the advantage of obtaining MSCs quickly and easily, and all processes from harvesting to transplantation can be performed in a single operation [41]. Considering the cell concentration, the most commonly used method for collecting BMAC is to collect aspirate from the iliac crest [42], but BMAC can also be collected from other sites, such as the proximal tibia [43]. Because only approximately 0.001% of MSCs of nucleated cells are obtained from aspirate, they are concentrated using density-gradient centrifugation to increase the number [44]. The concentrated BMAC is then injected directly into the affected joint or used in combination with other surgical procedures, such as proximal tibial osteotomy, to promote cartilage regeneration. Various growth factors included in BMAC induce cartilage regeneration and provide a favorable environment for MSC adhesion [45]. The immune control and anti-inflammatory effects of BMAC also help restore cartilage [40]. In patients receiving HTO, microfractures treated with BMAC were reported to have better arthroscopic findings regarding cartilage recovery than microfractures not treated with BMAC [34]; however, limitations exist. Because of the systems used in the field, including centrifuges, obtaining uniform cell numbers and concentrations is difficult. In addition, the amount of ideal BMAC required per unit defect size has not been established [46].

hUCB-MSCs are mesenchymal stem cells that are derived from human umbilical cord blood. Recently, hUCB-MSCs have been selected as a new treatment option for cartilage regeneration, and improved clinical results of knee OA with application of hUCB-MSCs have been reported in several studies [47,48]. These cells have been shown to have a high potential for cartilage regeneration, as well as anti-inflammatory and immunomodulatory properties [36,49]. hUCB-MSCs can be obtained from a donor, and they are typically expanded in culture before being injected directly into the affected joint or used in combination with other surgical procedures. hUCB-MSCs have several advantages over BMAC. First, hUCB-MSC showed higher proliferation rates and more than 1000-fold greater expansion capacity [49]. Second, because hUCB-MSCs are easy to obtain through cord blood banks, donor site morbidity is not a concern. However, the average additional cost of approximately $5300 or more compared with BMAC is a major obstacle to hUCB-MSC treatment.

The specific surgical method for using BMAC or hUCB-MSCs in combination with proximal tibial osteotomy will depend on the individual patient’s condition and the surgeon’s preferences. However, in general, these procedures involve the transplantation of stem cells with cartilage preparations such as microfracture before or after osteotomy [10].

Various treatment options that can be combined with HTO for cartridge regeneration have been reviewed; however, direct comparisons between BMAC and hUCB-MSCs are limited. To the best of our knowledge, this study is the first meta-analysis of the clinical outcome of HTO combined with BMAC or hUCB-MSCs. In the present study, both groups showed significantly improved postoperative clinical outcomes compared with preoperative baseline measurements. Basically, HTO results in clinical improvement because the joint reaction force is transferred to the lateral side of the knee via mechanical axis transfer, which is located lateral to the intercondylar eminence or at least to the center of the knee [50]. In a meta-analysis of seven studies to confirm the effectiveness of concurrent procedures during HTO [10], the combined cartilage regeneration procedure reportedly had a slightly beneficial effect on clinical outcome. However, in subgroup analysis of that meta-analysis, similar to the results obtained in the present study, the clinical results were significantly improved in the MSC subgroup. These positive synergistic effects are likely a result of the more effective cartilage regeneration potential and pain-reducing anti-inflammatory properties of MSCs due to paracrine effects. Magnetic resonance imaging (MRI) and histological examinations have been used to evaluate the improvement in cartilage status. MRI or histological examination was not used in any of the seven studies reviewed in the present systematic analysis. However, improvements in cartilage condition were observed in four studies [23,24,34,37,38] in which comparable preoperative and postoperative ICRS grades were reported, and hUCB-MSCs were significantly superior to BMAC in the meta-analysis performed.

The potential side effects associated with MSC-based treatments include local reactions, such as pain and swelling, as well as the potential for cells to differentiate into inappropriate cell types and the risk of tumor growth [51]. However, tumor growth is not a common side effect of MSC-based treatments [52]. Among the studies included in this analysis, swelling was the most common adverse event, and no adverse events met the criteria for being classified as serious according to the individual study protocols. However, the follow-up period in these studies was not sufficiently long to assess the risk of tumorigenicity.

The current review is the first systematic review and meta-analysis comparing the results of HTO combined with BMAC or hUCB-MSCs. However, the present review had several limitations. First, the studies included in the review were of a retrospective nature with levels of evidence of 3 or 4. Due to the absence of randomized controlled trials and prospective comparative studies, further studies with higher power are needed to control for possible confounding factors and to strengthen or refute current conclusions. However, due to the inherent strength of meta-analyses, an analysis including the single-arm study was conducted to provide useful information to clinicians. Second, sensitivity analysis confirmed high heterogeneity between studies, for which a random effects model was adopted when integrating the results. In addition, an independent analysis on various scoring scales used to evaluate clinical results was performed to accommodate the diversity of outcome variables. Third, the follow-up period of the reviewed studies was not sufficiently long to evaluate long-term clinical outcomes and survival rates. The follow-up period is an important factor in evaluating the stability of orthobiologics, so the results of this study should be interpreted carefully. In addition, the healing and maturation processes of articular cartilage may vary depending on the timing of second-look arthroscopy, resulting in potential bias. Finally, the absence of a control group that underwent HTO alone may bias the interpretation of the results.

In future studies, it is recommended to consider standardized outcome measures for assessing clinical improvement, as well as imaging techniques such as MRI or histological examination to evaluate the quality and quantity of cartilage regeneration.

## 5. Conclusions

Clinical outcomes were improved for both BMAC and hUCB-MSCs combined with HTO in varus knee OA patients, and postoperative outcomes did not differ between the two groups. However, hUCB-MSCs showed better articular cartilage regeneration in ICRS-CRA grade compared with BMAC. Further verification of the results requires larger, well-designed randomized controlled trials that include assessments of long-term follow-up.

## Figures and Tables

**Figure 1 medicina-59-00634-f001:**
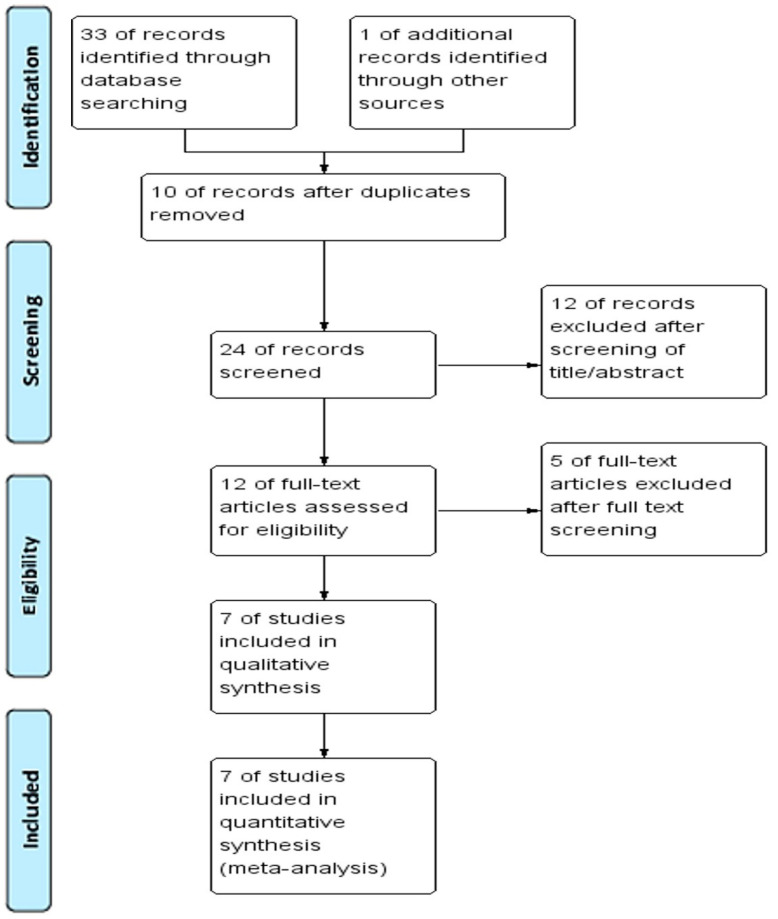
PRISMA flow diagram for the systematic review.

**Figure 2 medicina-59-00634-f002:**
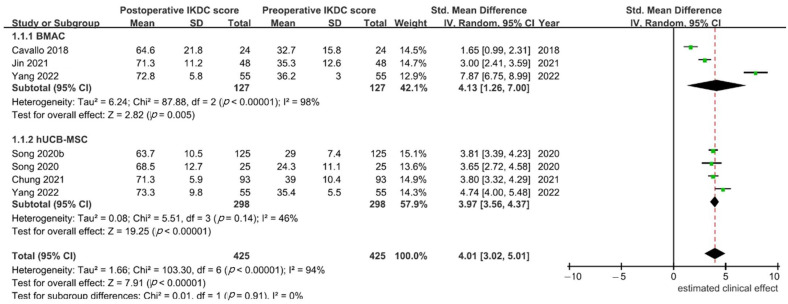
Estimated clinical effects of bone marrow aspirate concentrate (BMAC) and human umbilical cord blood-derived mesenchymal stem cells (hUCB-MSCs) in knee osteoarthritis (OA) patients treated with high tibial osteotomy (HTO) based on the International Knee Documentation Committee (IKDC) score [23,34,35,36,37,38].

**Figure 3 medicina-59-00634-f003:**
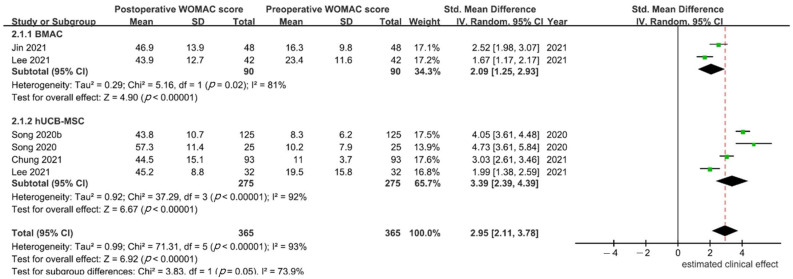
Estimated clinical effects of bone marrow aspirate concentrate (BMAC) and human umbilical cord blood-derived mesenchymal stem cells (hUCB-MSCs) in knee osteoarthritis (OA) patients treated with high tibial osteotomy (HTO) based on the Western Ontario and McMaster Universities OA Index (WOMAC) [24,34,36,37,38].

**Figure 4 medicina-59-00634-f004:**
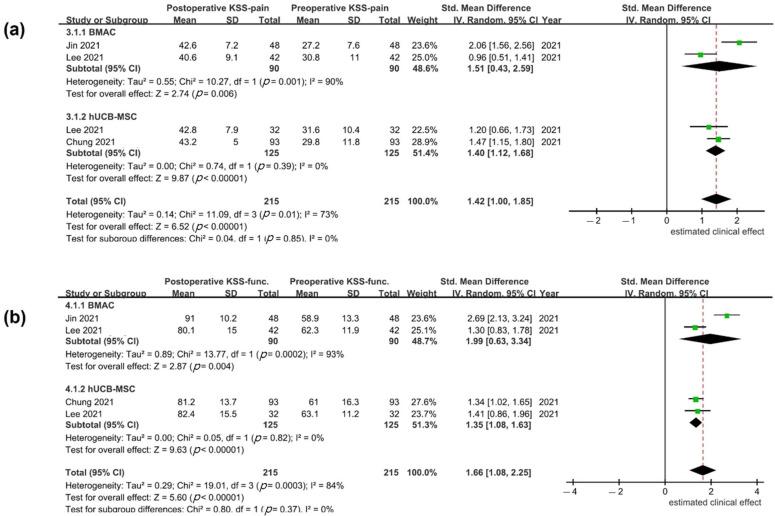
Estimated clinical effects of bone marrow aspirate concentrate (BMAC) and human umbilical cord blood-derived mesenchymal stem cells (hUCB-MSCs) in knee osteoarthritis (OA) patients treated with high tibial osteotomy (HTO) based on Knee Society Scores of (**a**) pain (KSS-pain) and (**b**) function (KSS-function) [24,34,38].

**Figure 5 medicina-59-00634-f005:**
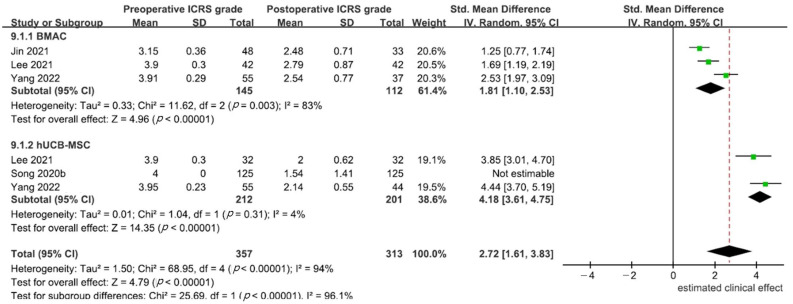
Estimated clinical effects of bone marrow aspirate concentrate (BMAC) and human umbilical cord blood-derived mesenchymal stem cells (hUCB-MSCs) in knee osteoarthritis (OA) patients treated with high tibial osteotomy (HTO) based on second-look arthroscopic articular cartilage status [23,24,34,37].

**Table 4 medicina-59-00634-t004:** Reported clinical outcome measures of the included studies.

Study	Reported Outcomes	IKDC	WOMAC	KSS-Pain	KSS-Function	KOOS	VAS	HSS	SF-36	Tegner
Cavallo, 2018 [35]	IKDC, KOOS, VAS, Tegner	BMAC, 32.7 ± 15.8 vs. 64.6 ± 21.8	-	-	-	BMAC, 30.46 ± 11.67 vs. 72.38 ± 20.1	BMAC, 7.50 ± 1.24 vs. 3.00 ± 2.08	-	-	BMAC, 1.21 ± 1.02 vs. 2.12 ± 1.39
Song, 2020 [36]	IKDC, VAS, WOMAC	hUCB-MSC, 24.3 ± 11.1 vs. 68.5 ± 12.7	hUCB-MSC, 57.3 ± 11.4 vs. 10.2 ± 7.9	-	-	-	hUCB-MSC, 76.4 ± 16.6 vs. 12.8 ± 11.7	-	-	-
Song, 2020 [37]	IKDC, WOMAC, VAS	hUCB-MSC, 29 ± 7.4 vs. 63.7 ± 10.5	hUCB-MSC, 43.8 ± 10.7 vs. 8.3 ± 6.2	-	-	-	hUCB-MSC, 7.6 ± 1.36 vs. 1.7 ± 1.4	-	-	-
Jin, 2021 [34]	IKDC, WOMAC, KSS-pain, KSS-function	BMAC, 35.3 ± 12.6 vs. 71.3 ± 11.2	BMAC, 46.9 ± 13.9 vs. 16.3 ± 9.8	BMAC, 27.2 ± 7.6 vs. 42.6 ± 7.2	BMAC, 58.9 ± 13.3 vs. 91.0 ± 10.2	-	-	-	-	-
Chung, 2021 [38]	IKDC, WOMAC, KSS-pain, KSS-function, HSS	hUCB-MSC, 39.0 ± 10.4 vs. 71.3 ± 5.9	hUCB-MSC, 44.5 ± 15.1 vs. 11.0 ± 3.7	hUCB-MSC, 29.8 ± 11.8 vs. 43.2 ± 5.0	hUCB-MSC, 61.0 ± 16.3 vs. 81.2 ± 13.7	-	-	hUCB-MSC, 61.6 ± 12.9 vs. 82.7 ± 13.5	-	-
Lee, 2021 [24]	HSS, WOMAC, KSS-pain, KSS-function	-	BMAC, 43.9 ± 12.7 vs. 23.4 ± 11.6; hUCB-MSC, 45.2 ± 8.8 vs. 19.5 ± 15.8	BMAC, 30.8 ± 11.0 vs. 40.6 ± 6.1; hUCB-MSC, 31.6 ± 10.4 vs. 42.8 ± 7.9	BMAC, 62.3 ± 11.9 vs. 80.1 ± 15.0; hUCB-MSC, 63.1 ± 11.2 vs. 82.4 ± 15.5	-	-	BMAC, 57.9 ± 12.9 vs. 79.2 ± 11.5; hUCB-MSC, 56.1 ± 10.6 vs. 84.6 ± 15.5	-	-
Yang, 2022 [23]	IKDC, KOOS, SF-36, Tegner	BMAC, 43.9 ± 12.7 vs. 23.4 ± 11.6; hUCB-MSC, 45.2 ± 8.8 vs. 19.5 ± 15.8	-	-	-	BMAC, 37.7 ± 2.7 vs. 78.2 ± 7.9; hUCB-MSC, 36.8 ± 7.1 vs. 78.4 ± 8.6	-	-	BMAC, 43.6 ± 10.4 vs. 64.0 ± 11.6; hUCB-MSC, 45.8 ± 12.3 vs. 64.5 ± 11.9	BMAC, 2.3 ± 0.9 vs. 4.0 ± 0.5; hUCB-MSC, 2.2 ± 0.8 vs. 4.1 ± 0.5

IKDC, International Knee Documentation Committee; KOOS, Knee injury and Osteoarthritis Outcome Score; VAS, Visual Analog Scale; WOMAC, Western Ontario and McMaster Universities Arthritis index; KSS, Knee Society Score; HSS, Hospital for Special Surgery; SF-36, Short Form 36 Health Survey.

## Data Availability

The data published in this research are available on request from the first author (D.P.).

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
