# Peer review of "Bone Marrow Aspirate Concentrate versus Human Umbilical Cord Blood-Derived Mesenchymal Stem Cells for Combined Cartilage Regeneration Procedure in Patients Undergoing High Tibial Osteotomy: A Systematic Review and Meta-Analysis"

_medicina, 2023, doi:10.3390/medicina59030634_

Round 1
Reviewer 1 Report
Dear Authors,
Thank you for this interesting study. I think that despite the low quality of evidence involved, this study is important and provides insight into the role of orthobiologics in OA treatment.
I do have a few comments mainly for the discussion section.
1. line 270-272: Several studies have reported successful short- and medi- um-term outcomes after HTO [6, 7]; however, long-term outcomes were inferior to those prior [8, 9]. - unclear sentence, to which prior are the authors relating?
2. line 272 - incorrect use of "is" - are associated
3. line 285 - BMAC can be collected from several areas, the most popular are iliac crest and proximal tibia. in the context of knee surgery proximal tibia is more convenient, though, recent study showed slightly lower cell concentration compared to iliac crest aspirate. While this discussion is out of the scope of the article, it should be mentioned that the iliac crest is not the only origin of BMAC.
4. line 302-303 - lower risk of complications from post-transplantation immunological reactions - lower risk compared to what? BMAC is autologous therefore immune reactions are not a concern at all.
5. line 312 - laterally distributed on the medial side - mechanical axis is transferred lateral to the intercondylar eminence (or at least to the center of the knee), therefore, this sentence is inaccurate.
6. lines 326-330 - paragraph starting with The common side effects - the first sentence implies that common side effects include tumor growth, which is not true. when mentioning that no serious adverse events were observed, what is considered to be a serious adverse event should be clarified.
Reviewer 2 Report
The manuscript entiteled: Bone Marrow Aspirate Concentrate versus Human Umbilical Cord Blood-Derived Mesenchymal Stem Cells for Combined Cartilage Regeneration Procedure in Patients Undergoing High Tibial Osteotomy: A Systematic Review and Meta-Analysis describes the outcome of the application of two therapeutically measures for tissue regeneration.
It is a valuable assessment of the available study performed and their reliable descriptions. The manuscript is well structured and informative.
Although I would appreciate a more detailed information on the two methods and the surgery procedure itself. What is their importance in the whole concert of options for intervention? How relevant and frequent are the compared techniques.
Furthermore, I miss a condensed suggestion for the next study reports. Which information should be there, what is necessary to improve the informative value and to equalize the outcome. It is somewhere distributed in the text but a final list or table would be good. O a general recommendation scheme about the parameters to be measured and the follow up period including necessary diagnostics and imaging techniques could be added.
I have only a few minor comments.
Page2 line 82: start date of the search?
Page 3 line 96: how many cases of uncertainty
Page 4 table Chung 21 – calculate the month not the years as for the others
Figure 2-5 – to small
Discussion line 260: various ?
